Oceanographic moorings as year-round laboratories for investigating growth performance and settlement dynamics in the Antarctic scallop Adamussium colbecki (E. A. Smith, 1902)

http://orcid.org/0000-0002-0137-3605 Schiaparelli Stefano 1 2 stefano.schiaparelli@unige.it
http://orcid.org/0000-0002-2555-4398 Aliani Stefano 3
1 DISTAV, University of Genoa , Genoa , Italy
2 Italian National Antarctic Museum (MNA, Section of Genoa), University of Genoa , Genoa , Italy
3 Istituto di Scienze Marine, Italian National Research Council (CNR) , La Spezia , Italy
Heron Scott
Electronic publication date: 2019 Mar 21
Publication date: 2019
Volume: 7
Electronic Location ID: e6373
Received 2018 Mar 19; Accepted 2018 Dec 30
Copyright: © 2019 Schiaparelli and Aliani
Copyright year: 2019
Copyright holder: Schiaparelli and Aliani
License: This is an open access article distributed under the terms of the Creative Commons Attribution License, which permits unrestricted use, distribution, reproduction and adaptation in any medium and for any purpose provided that it is properly attributed. For attribution, the original author(s), title, publication source (PeerJ) and either DOI or URL of the article must be cited.
License URL: https://creativecommons.org/licenses/by/4.0/

Keywords: Settlement, Time-series, Monitoring, Caging, Adamussium colbecki, Mooring, Growth, Ross sea, Antarctica

Funding: Italian National Scientific Commission for Antarctic Research (CSNA) Participation of Stefano Schiaparelli to the SOOS Ross Sea working group meeting has been funded by the Italian National Scientific Commission for Antarctic Research (CSNA). The funders had no role in study design, data collection and analysis, decision to publish, or preparation of the manuscript.

==============================
Background

Oceanographic moorings (OMs) are standard marine platforms composed of wires, buoys, weights and instruments, and are used as in situ observatories to record water column properties. However, OMs are also comprised of hard substrates on which a variety of invertebrates can settle when they encounter these structures along their dispersal routes. In this contribution, we studied the fouling communities found on two OMs deployed in the Ross Sea (Antarctica). Furthermore, a cage containing the Antarctic scallop Adamussium colbecki (E. A. Smith, 1902) was incorporated in the OM. The growth of the caged A. colbecki were evaluated after 1 year and their shells used as biological proxy for seawater temperature and salinity.

Methods

A variety of settlers were collected from two different OMs deployed in the Ross Sea (Antarctica) and species identified using a combination of morphological and genetic (mainly through DNA barcoding) characteristics. Caged scallops were individually marked with permanent tags and their growth studied in terms of size-increment data (SID). Cages were specifically designed to prevent damage to individuals due to water drag during OM deployment and retrieval. Growth parameters from the caged individuals were applied to the A. colbecki juveniles that had settled on the mooring, to trace the likely settlement period.

Results

The growth performance of caged A. colbecki was similar to that from previous growth studies of this species. The remarkable survival rate of caged specimens (96.6%) supports the feasibility of caging experiments, even for a species with a fragile shell such as the Antarctic scallop. Some of the new recruits found on the mooring were A. colbecki, the same species we put into special cages fixed to it. The settlement of the A. colbecki juveniles started during the Austral spring with a peak in summer months and, remarkably, coincided with seasonal changes in water temperature and flow direction, which were recorded by the mooring’s instruments. Genetic data from other settlers provided new information about their larval ecology and connectivity.

Discussion

Oceanographic moorings are expensive and complex experimental platforms that, at present, are strictly used for the acquisition of physical and biogeochemical data. Their use for in situ ecological experiments on model organisms suitable for caging and to study fouling species has yet to be fully explored. We present the outcomes of a study, which represents a baseline for the characterization of Antarctic fouling biodiversity. We hope that in the near future an internationally coordinated systematic study of settlers could be initiated around the Antarctic continent. This could utilize “new generation OMs” equipped with standardized settlement structures and agreed sampling protocols for the study of fouling communities.

Introduction

Biological monitoring, or biomonitoring, is based on the use of model species or “bioindicators” to assess changes in the environment. Candidate species for such activities are usually organisms with well-studied ecologies and that are “robust enough” to survive in different environmental settings. A significative number of individuals must survive manipulation during initial sampling and subsequent translocation to the in situ study site. Many invertebrate species with these characteristics have been widely used to detect presence and concentrations of pollutants in the environment, with filter-feeder bivalves such as mussels and oysters being amongst the most suitable marine species (Fortunato, 2015). Some monitoring activities centered on bivalves now have a long history, such as the U.S. Mussel watch program which started more than 30 years ago (Goldberg et al., 1978; https://catalog.data.gov/dataset/national-status-and-trends-mussel-watch-program), and which has enabled monitoring at large geographical scales, that is, spanning thousands of kilometers.

Whenever an indicator species is not naturally present in the area to be monitored, biomonitoring studies have involved translocating specimens from a source population (which is usually also used as control site) to the impacted area. This technique is defined as “active bio-monitoring” (De Kock & Kramer, 1994). The in situ exposure of translocated bivalves usually spans from 1 to 3 months, which represents the amount of time necessary for them to accumulate measurable quantities of pollutants (Roméo et al., 2003; Andral et al., 2004). The bivalves are contained in submerged cages, generally positioned in shallow waters (1–10 m depth) (Andral, Galgani & Blottiere, 2007) by divers and eventually retrieved by grapple hooks (Roméo et al., 2003; Gorbi et al., 2008). This “caging approach” can also be used in deep waters, with cages attached to lines moored with weights and stabilized by mid-water and surface buoys. For example, specimens of Mytilus galloprovincialis Lamarck, 1819 were successfully suspended in cages in 40–1,550 m (Galgani et al., 2005). This species tolerated the initial high-speed immersion (120 m min−1), with 38% surviving to the end of the experiment (Galgani et al., 2005). However, cages attached to surface lines and buoys represent a potential navigation hazard, especially when deployed to monitor deep-water discharges or oil fields, which are often located along commercial navigation routes. Surface buoys also cannot be used on a freezing sea surface, if the study is performed at high latitudes. Acoustic release technology allows retrieval of baited traps or cages without the need for any additional physical structure floating at the water surface. With this technique, however, the organisms targeted for biomonitoring are generally not chosen “a priori” but obtained from the pool of resident species, usually deep-water fishes and amphipods (Sundt, Børseth & Myhre, 2008).

In a completely different research context, oceanographic moorings (OMs), that is, marine platforms composed of wires, buoys, weights and instruments, are deployed in selected areas as in situ observatories to measure and record water column oceanographic properties. Thanks to the wide variety of scientific instruments and sensors fixed along the array, OMs collect time-series measurements of a wide range of variables (e.g., current direction and velocity, temperature, conductivity, turbidity, photosynthetically active radiation, chlorophyll, hydrocarbons, methane concentrations, intensity of particulate fluxes). Large sediment traps are also commonly fixed on such moorings. These OMs also intrinsically represent simple hard substrates on which a variety of invertebrates may settle. For example, a community of species dispersing over long distances in the ocean, which can be included in the widely spread “Club of Superwanderers” (Cornelius, 1992; Aliani & Meloni, 1999), often settle on these offshore structures.

Under the Italian Programma Nazionale di Ricerche in Antartide (PNRA) research project “Polar DOVE” (“Variability of abyssal polar ventilation and its impact on the global circulation,” PNRA 2004/8.2), we trialed the addition of cages containing a model organism to existing OM arrays. One year later, we sampled the organisms that had settled on the hard structures of the OM (instruments and buoys) and on the surfaces of the cages. The main aim of the “Polar DOVE” project was to study trace element ratios and isotopic composition in skeletal elements or shells of invertebrates, as a proxy for records of environmental temperature and salinity. This project targeted Antarctic species with “robust” carbonate structures with proper elemental and biological features. Data on trace elements and isotopic composition obtained from their skeletons would then able to be calibrated against and compared to data collected from sensors on the OM array.

The species initially targeted were the barnacle Bathylasma corolliforme (Hoek, 1883), the cup coral Flabellum spp., and the Antarctic scallop Adamussium colbecki (E. A. Smith, 1902). The latter species was eventually chosen as most appropriate, given: (i) its abundance in the study area (Terra Nova Bay (TNB), Ross Sea), especially at diving depths; (ii) its trophic flexibility (it is capable of feeding on resuspended particulate organic matter, as well as on phytoplankton; Norkko et al., 2007); and (iii) the fact that A. colbecki spat have been repeatedly found on mooring structures in the past (authors unpublished observations).

In this paper, we present the methods used in the “Polar DOVE” experiment. An OM was used to support cages containing an Antarctic “model species,” which served as a “bioindicator” of water column properties. Additionally, we discuss the use of this OM structure to investigate settlement dynamics of Antarctic invertebrates, including those of the “model species.” The calibration of the biological proxy with instrumental data has been reported elsewhere (Trevisiol et al., 2013). Our final aim here is to introduce the potential use of OMs as unique multidisciplinary platforms to study settlement dynamics of Antarctic invertebrates and to perform cage experiments.

Material and Methods

Study site

All the field activities were performed in 2005–2006 during the XXI PNRA Italian Antarctic expedition at TNB in the Ross Sea (Fig. 1). The mooring with the cages was placed in the Adelie Cove area, north of Hell’s Gate and close to the Italian Antarctic station “Mario Zucchelli,” at the limit of TNB Antarctic specially protected area n. 161. Adelie Cove is a “V-shaped” bay about 70-m deep, which is subjected to katabatic winds that can drive bentho-pelagic processes (Povero et al., 2001).

Figure 1 Map of Terra Nova Bay with the position of the mooring “L,” close to the Adélie penguin rookery.

The map was produced with Quantarctica V3 (http://quantarctica.npolar.no/about.html) using the satellite layer Landsat Image Mosaic of Antarctica (LIMA) (U.S. Geological Survey, USGS- Bindschadler et al., 2008) which is in the public domain. Landsat Image Mosaic of Antarctica (LIMA) courtesy of the U.S. Geological Survey.

Mooring and cage structure

The mooring placed at Adelie Cove is coded as “Mooring L” within the PNRA mooring program (http://www.soos.aq/news/current-news/202-italian-moorings) and was a standard oceanographic array about 100 m long, which included: (i) an upper floatation buoy at about 45 m of depth, (ii) one Aanderaa RCM7 current meter, (iii) a SeaBird SBE 37 temperature recorder immediately below, and (iv) two in situ cages containing A. colbecki (see Fig. 2 for details). The Aanderaa model RCM7 has one cm s−1 accuracy in speed and five degrees in direction, and the SBE 37 a 0.002 °C temperature accuracy. All instruments were factory-calibrated, a post-deployment check was also performed, and the data checked for errors (Trevisiol et al., 2013).

Figure 2 The mooring “L” array.

The two cages designed to hold the A. colbecki specimens were constructed from modified Polyvinyl chloride (PVC) pipe designed for high-pressure underground discharge conduits (UNI EN 1401) (Fig. 3). Holes were made in the pipe to create a frame with large openings (Fig. 3A), within which were placed mesh “lanterns” (as used for oyster farming; diameter = 38 cm, plastic net mesh size = one cm) to house the scallops. These lanterns had three intermediate internal separations or “floors.” The PVC structure protected the lanterns from collapsing and from any external damage. This design ensured good water flow through the cages and hence food supply to the scallops. Handcrafted, seawater-resistant stainless steel frames and clamps were used to fix cages to the Kevlar mooring rope (Fig. 3B). Sacrificial anodes placed on the steel frames protected the metal against corrosion (Fig. 3C). Two plastic funnels were also fixed to the top and bottom of the PVC tubes (Fig. 3A) in order to: (i) reduce hydrodynamic forces during mooring deployment and recovery and (ii) reduce potential disturbance to specimens from sedimentation. Each lantern floor hosted 10 A. colbecki specimens, with a total of 30 bivalves per cage. Cages were placed at 54 (“Cage 1”) and 59 (“Cage 2”) m depth. The closest temperature sensor was at 52 m and the current meter was located at about 10 m above “Cage 2.” Mooring “L,” with the cages, was deployed on January 16th, 2006 (at 14:14 UTC) at a bottom depth of 145 m (GPS coordinates 74°44.6048′S 164°8.3916′E) and recovered on January 31st, 2007, after 380 days at sea.

Figure 3 Cages’ structure Cage structure and elements prior to deployment.

(A) The high-pressure PVC tube with the windows to enable water-carried particles to reach the lantern. The two plastic funnels decrease drag and turbulence during mooring deployment and retrieval and are kept in place by electric cables. The green ropes protruding from the funnels are the cage ropes, which were secured and fastened to the mooring rope (red). (B) A close view of one of the stainless steel clamps which enabled the cage to be attached to the mooring rope. (C) The closure system (opposite side of clamps) of the stainless steel frame; the small cylinder on the left is a sacrificial anode.

Adamussium colbecki tagging

A set of 60 undamaged A. colbecki specimens (i.e., with perfect shells and without encrustations) were collected by divers at 25 m depth at Tethys Bay (close to “Mario Zucchelli” station) to be caged on the mooring. All specimens were permanently marked with bee-tags (“Opalith-plättche”; IMKEREI manufacturer, Dieter Rudolph, Bremen-Oberneuland) on both valves. Tags were glued to the shells, about one cm from the shell edge, using epoxy resin. Three tags were attached to each valve to maximize the possibility of specimen identification in case of heavy shell breakage. Large A. colbecki specimens were deliberately chosen (average shell length = 66.24 ± 5.58 mm, n = 60) to minimize the potential breakage of shells during mooring deployment (likely to have been more frequent in thinner shelled juvenile specimens). A total of 36 of the tagged A. colbecki (those not used by Trevisiol et al. (2013)) are retained in the Italian National Antarctic Museum collections (MNA, Section of Genoa) under the vouchers MNA 9162–MNA 9197. Specimen collections and field activities were conducted under fieldwork permit approval of the Italian National Antarctic Program (PNRA), signed on January 12th, 2006, and complied with the “Protocol on Environmental Protection to the Antarctic Treaty” (Annex II, Art. 3).

Caged Adamussium colbecki growth performance

In this tag-recapture experiment, Size-Increment Data (SID) were calculated for each A. colbecki specimen at t0 (time of deployment) and t1 (time of retrieval), by measuring shell length (height, umbo to edge of shell along axis of growth) of the upper valve. Measurements were made using a Vernier callipers (precision of 0.01 mm). These data were used to estimate von Bertalanffy function parameters by applying: (i) the Walford method (Walford, 1946) and (ii) a non-linear iterative fitting NEWTON algorithm (Fabens, 1965; Brey, 2001). The Walford method uses a linearizing transformation of the growth function to estimate the parameters L∞, the asymptotic shell length (height) and K, the Brody growth coefficient of the von Bertalanffy growth function: Lt1= L∞(1−e−K(t1−t0))

where Lt1 is the length at time t1. The linear transformation is obtained through the regression: Lt1= a + bLt0

which allows the calculation of L∞, corresponding to a/(1−b), and of K, which is equal to: −ln(b).

For the iterative fitting, a von Bertalanffy specialized model was used (Fabens, 1965): Lt1= Lt0+ (L∞− Lt0)(1−e−Kdt)

Simple methods for estimating von Bertalanffy growth coefficients, such as the Ford–Walford plots or Fabens’ method, may lead to biased estimates (Hart & Chute, 2009 and references therein) due to the fact that: (i) growth parameters may vary among individuals and (ii) the available specimens in a study may not cover the whole range of sizes present in a population. The latter was true in our study, as we deployed larger specimens. To partially accommodate this fact, and because our A. colbecki were collected from the same location as a population studied by Chiantore, Cattaneo-Vietti & Heilmayer (2003), we also followed Chiantore, Cattaneo-Vietti & Heilmayer (2003) in substituting our estimated L∞ of ∼81 mm (see results), with 92 mm, the known maximum size reached by this species in the area (Fig. S1).

Adamussium colbecki settlement and study of other fouling organisms

The newly settled A. colbecki, were measured using a stereomicroscope equipped with a scale, and size-frequency distributions of shell lengths (height, umbo to edge of shell along axis of growth) were calculated considering size class increments of 0.5 mm. Size-frequencies were then grouped into five cohorts bins (corresponding to the sizes of five to six, four to five, three to four, two to three and one to two mm) to be modelled using the R (R Core Team, 2018) package “mixdist” (MacDonald & Du, 2018, version 0.5-4). Mixdist offers combined algorithms (i.e., Expectation-Maximization and Newton-type method) for fitting finite mixture distributions and evaluate the presence of a multimodal distribution, which, in the case of size-frequency data, would indicate the existence of different cohorts. The fitting of the model was run with ten EMs. Mixdist output plot reports the single multiple distributions in red, with the modes indicates by triangles and their sum shown as a thick green line. At the end of the mixture distribution analysis the thick green line should match the shape of the histogram as closely as possible. A Goodness-of-fit (chi-squared ANOVA), normal distributions and best fitting constraints were also used following Nygård, Vihtakari & Berge (2009). The A. colbecki juveniles found on the funnels and measured for the present study are deposited in the MNA collections under the single lot MNA 9161. Other invertebrates found on this mooring (“L”; also from other years) and on mooring “D” (Fig. S2) were identified taxonomically and, in some cases, also using molecular techniques (see below) (Table S1; Fig. S3).

Molecular analyses

DNA extraction, amplification (with primers LCOech1aF1 and HCO2198, Folmer et al., 1994) and sequencing of partial cytochrome c oxidase subunit 1 (CO1) for crinoids and polychaetes were carried out at the Canadian Centre for DNA Barcoding (University of Guelph, Ontario, Canada). For the gastropod Capulus subcompressus Pelseneer, 1903, DNA extraction was done with phenol–chloroform (see Fassio et al., 2015 for details) and two genes were sequenced: (i) the cytochrome oxidase I gene (COI) (with primers: LCO1490 and HC02198, Folmer et al., 1994) and (ii) the 16S rDNA (with primers 16SA, Palumbi, 1996 and CGLeuR, Hayashi, 2003). All sequences were uploaded on the BOLD platform (Barcode of Life Data systems, http://www.boldsystems.org) and on Genbank (Table S1). Genetic analyses were performed in the framework the PNRA program Barcoding of Antarctic Marine Biodiversity (“BAMBi,” PNRA2010/A1.10).

Timeseries (temperature and currents)

Temperature data were collected from Adelie Cove in two different years using the Aanderaa current meter and CTD recorders from the SBE. Data were merged in a continuous series from January 1, 2005 to January 2007. Water current data were available for the period of the caging experiment only due to a technical failure in the current meter deployed the previous year. Therefore, the dataset available for this study encompasses the years 2005–2007 for temperature and 2006–2007 for currents and cages.

The vertical structure of the water column in the wider TNB is relatively simple with the greatest variability being in the surface layer that extends to 50–150 m depth. In wintertime the prevalent water mass is High Salinity Shelf Water (HSSW) with a temperature at the surface freezing point. In summertime, the Summer Surface Water (SSW) occupies the upper layer (Assmann, Hellmer & Beckmann, 2003) that becomes fresher and warmer because it is influenced by summer sea-ice melting and by the heat gain from solar radiation (Kern & Aliani, 2011). We have plotted seawater temperatures in blue when below −1.7 °C (i.e., when the sea ice is still present) and in red when warmer than −1.7 °C (i.e., when the trend is inverted and the sea ice is starting to melt).

Results

Adamussium colbecki survivorship and growth performance

Survivorship of A. colbecki at the end of the 380 days was extremely high, at 96.6%. A total of 58 of the 60 individuals added to the cages were healthy and, of these, only six specimens were excluded from SID calculations because their shell edges were too damaged. The two dead specimens were found as loose valves (i.e., with valves not connected by the ligament) and possibly died shortly after mooring deployment.

Growth parameters, obtained by applying the Walford method (Fig. 4) and the fitting procedure with the Newton algorithm produced almost identical data with, respectively, L∞ = 80.92617, K = 0.259937 and L∞ = 80.92, K = 0.26 values. These parameters defined von Bertalanffy growth functions (Fig. 5), which lie in a cluster of curves obtained in the framework of different studies modelling A. colbecki growth (see Heilmayer et al., 2003 for details on A. colbecki growth performances). K values in our data (i.e., ∼0.26) are higher than those obtained in the majority of previous studies (which ranged from 0.11 to 0.19; see Heilmayer et al., 2003). Our K values are closer, at least in the first part of the growth curve (up to 5 years of age), to those of Ralph & Maxwell (1977) who obtained a K = 0.24 based on X-rays examination of growth rings. In Fig. S1 we report our growth curves with a modified L∞ value, that is, set to 92 mm, using the known maximum size attained by this species in the same area (Chiantore, Cattaneo-Vietti & Heilmayer, 2003). With this artificially set L∞, our growth curves fit those of Ralph & Maxwell (1977) at all ages (Fig. S1).

Figure 4 Walford plot regression equation.

Lt1=Lt0*0.771+18.524; R2 = 0.869. Gray area represents 95% confidence intervals.

Figure 5 Von Bertalanffy growth functions from present and literature data.

Timeseries

Water currents were in two prevalent directions, toward southwest and toward northeast all year long (Fig. 6A). From January to June 2006, southwest currents were more frequent, although pulses in opposite directions were also recorded. The average current speed during this time period was 13 ± 9 cm/s, with maximum up to 36 cm/s. From June to early October velocities were lower (mean 11 ± 5 cm/s) but many days with minimum speeds of 1.4 cm/s were also recorded, especially in the last part of the deployment period. From late October to mid-December, currents were mainly toward a northerly direction and, while the average speed was lower (8.8 ± 7 cm/s), strong currents (40 cm/s) were also recorded. SSW temperatures ranged from −1.91 to −0.1 °C in both years (Fig. 6B). The short gap visible in the temperature series (Fig. 6B) is due to the planned yearly mooring maintenance and corresponds to the start of the caging experiment.

Figure 6 Stick plot of current meter data, temperature and Adamussium settlement events.

Stick plot of current meter and temperature data. (A) Current direction and speed are colored according to seawater temperature after removing tides with a 24 h moving average filter. Blue indicates seawater temperature below −1.7 °C and red indicates water masses warmer than −1.7 °C. (B) Temperature data. (C) Adamussium cohorts time of settlement calculated on the base of the von Bertalanffy growth curve function parameters estimated from present data (uncorrected L∞, blue line) and after Chiantore, Cattaneo-Vietti & Heilmayer (2003) (red line). Regardless the Von Bertalanffy parameters applied, settlement peaks fall in summer months. For this plot, Adamussium cohorts have been divided into five size classes bins (respectively, from left to right: five to six; four to five; three to four; two to three; one to two mm) in both cases (i.e., red and blue line).

Adamussium colbecki settlement

A total of 394 A. colbecki juveniles were retrieved on the plastic surfaces of the funnels (Fig. 7A), with shell lengths ranging from 1.04 to 5.6 mm. As they were present only on the external surfaces, we are confident they were not produced from spawning of the caged specimens. By applying the von Bertalanffy growth function parameters (with an uncorrected L∞) to the sizes of the juveniles it was possible to trace the “age” of each of the cohorts and plot the purported time of settlement (Fig. 6C). The peak of settlement occurs in January–February, with the earliest A. colbecki settlers arriving in November (Fig. 6C, blue line). Settlement matches the seasonal temperature increase and the change in water flow direction (Figs. 6A and 6B). In order to complete the picture we have also considered different von Bertalanffy growth functions parameters, that is, those obtained by the only other study to model A. colbecki growth through SID data (Chiantore, Cattaneo-Vietti & Heilmayer, 2003) (Fig. 6C, red line). Regardless of how the von Bertalanffy parameters are applied, settlement peaks fall in summer months. If growth parameters with corrected L∞ (see Fig. S1) are used, settlement estimates using the two approaches are also very similar (2 days apart for the one to two mm cohort and 11 days apart for the five to six mm cohort). Measurements of A. colbecki juveniles revealed the presence of several cohorts but with a unimodal distribution of size frequencies (Fig. 8). The Goodness-of-fit (chi-squared ANOVA) indicates that the model estimated is consistent with the data (df = 6, Chi-sq. = 5.4093; P-value = 0.4925).

Figure 7 Most typical fouling organisms found on mooring “L”.

Most common fouling organisms found on mooring “L” (all images except E and H), and mooring “D” (images E and H). (A) Adamussium post-larvae (byssally attached) on the outer plastic surfaces of the funnels. (B) and (C) SEM images of Adamussium juveniles retrieved from the funnels. (D) Highlight of the Adamussium prodissoconch; the arrow marks the discontinuity between the prodissoconch I (diameter ∼110 μm) and the prodissoconch II (diameter ∼330 μm). (E–G) Serpulids attached to mooring buoys showing the ectoparasite mollusc Capulus subcompressus Pelseneer, 1903. (H) The barnacle Bathylasma corolliforme (Hoek, 1883) sampled on January 24th, 2012 on “mooring D” in the framework of the PNRA Project “BAMBi.” (I) Pentacrinoid larvae of Notocrinus virilis attached to the current meter case. Scale bars: (A) five mm; (B) one mm; (C) 500 μm; (D) 100 μm.

Figure 8 A. colbecki population structure of the newly recruited cohorts on the cage funnels.

Population structure of A. colbecki juveniles settled on the outer surface of the funnels. The final fitting finite mixture distribution is shown in green (see text for details).

Fouling characterization and molecular analyses

Several species belonging to the Phyla Cnidaria, Annelida, Arthropoda, Echinodermata, Mollusca and Chordata were found on the arrays of mooring “L” (Figs. 7E–7I; Table S1) and “D” (Figs. S2 and S3; Table S1). Molecular analyses enabled us to ascribe previously unidentified pentacrinoid larvae (MNA 9151; MNA 9153; MNA 9154; MNA 9159; MNA 9160; Fig. 7I) to Notocrinus virilis Mortensen, 1917 based on their COI sequences. The polynoid polychaetes sequenced belonged to Harmothoe fuligineum (Baird, 1865) (MNA 3403, Fig. S3F) and Harmothoe sp. (MNA 3415, Fig. S3H). Several specimens of Capulus subcompressus Pelseneer, 1903 were also found (Figs. 7E and 7F). The COI and 16S sequences of this gastropod have been described in Fassio et al. (2015), where a high genetic connectivity between the Ross Sea and Weddell Sea-Antarctic Peninsula was shown for this species.

Discussion

In the Southern Ocean, several national and international research programs currently maintain OM networks to understand the complex physical and chemical features of Antarctic water masses (a global view of recent projects using OMs can be found at: http://www.soos.aq/news/current-news/316-southern-ocean-mooring-data, list updated to 2017). However, these oceanographic structures have never been conceived as cross-disciplinary platforms to integrate physical and biological data, as proposed here. In our experiment a year-round OM, part of the Italian mooring observatory system in TNB, was used both to cage a locally ecologically important species, that is, the Antarctic scallop A. colbecki (and to test its utility as a biological proxy of seawater features), and to analyze settlement dynamics of this species. Both experiments have been accomplished with success. Despite their very thin shells, all but two of the caged A. colbecki (96.6%) survived the 380 days, as well as the deployment and retrieval procedures. The use of A. colbecki as a biological proxy is feasible and demonstrated that reconstructed temperatures obtained by studying the shells are closer to the mean summer temperature rather than the mean annual temperature (Trevisiol et al., 2013). Growth rates of our caged specimens (Fig. 5) were in line with what was already know for this species, the whole bunch of growth curves forming a “rather dense cluster of quite similar curves” (Heilmayer et al., 2003).

Adamussium colbecki has been the subject of research within the Italian PNRA since the first Italian expeditions, resulting in an unusual in depth knowledge for an Antarctic species, testified by a large number of published studies (Cerrano et al., 2000, 2001, 2006; Cerrano et al., 2009; Chiantore et al., 1998, 2000, 2001; Guidetti et al., 2006; Schiaparelli & Linse, 2006; Dell’Acqua et al., 2017; Moro et al., in press). This species was also proposed as a candidate for a “Southern Ocean mussel-watch program” (Berkman & Nigro, 1992; Nigro et al., 1997; Dalla Riva et al., 2004). The outcomes of the present study confirm its great potential in “biomonitoring” studies involving cages and in situ experiments. However, the “perception” of A. colbecki as a common species has been mainly driven by the high abundances registered in TNB and in few other Antarctic sites. Based on these data the species was erroneously believed to be widespread and abundant through all Antarctic coastal sites. That A. colbecki is in fact much rarer was only realized in 2004, when sampling during the NZ “BioRoss” voyage (Mitchell & Clark, 2004; Schiaparelli, Lörz & Cattaneo-Vietti, 2006) and the Italian “Latitudinal Gradient Project” voyage (Ghiglione et al., 2013; Schiaparelli et al., 2014) (http://www.lgp.aq) showed the almost complete absence of this species in other Ross Sea sites. Today, we know that A. colbecki occurs in locations characterized by great environmental stability, being absent or much less abundant in most Antarctic coastal areas (Berkman et al., 2004; Schiaparelli & Linse, 2006). Hence, the high abundances found in TNB are uncommon and due to a peculiar interplay of positive and locally important factors (Schiaparelli & Linse, 2006). Due to this updated distributional analysis, in 2009 A. colbecki was included in the CCAMLAR species list of VME indicators (https://www.ccamlr.org/en/system/files/e-sc-xxviii-a10.pdf). Therefore, the use of A. colbecki in biomonitoring activities, despite its “robustness” as “cage-species” proven here, should be carefully evaluated in the future, especially where this species is not naturally abundant, to avoid compromising the characteristics of existing populations.

Analysis of the population structure of the A. colbecki juveniles settling on the external surfaces of the funnels provided new insight on the timing of settlement in TNB. In fact, it remarkably coincided with a seasonal change in water direction (a switch from north to south east) and with an increase in seawater temperature (above −1.7 °C) that corresponds with the beginning of ice melt. The presence of warm water masses in the area remained constant from December to February 2007. Without details on other oceanographic processes, we assume that this environmental change, indicating that sea ice has melted and surface waters are being heated by increased solar radiation (Kern & Aliani, 2011; Assmann, Hellmer & Beckmann, 2003), also represents a relevant threshold for species growth and settlement. This information is rather important, as the timing of settlement of A. colbecki has long been debated in the literature, with contrasting situations and geographical mismatches in the purported timing of the spawning season (Chiantore et al., 2000; Heilmayer et al., 2003).

Adamussium colbecki has a 1-year gametogenic cycle, an unicum among Antarctic species (Tyler et al., 2003), after which unprotected planktotrophic larvae are produced (Berkman, Waller & Alexander, 1991). In southern McMurdo Sound (Southern Ross Sea, ∼78°S), spawning was reported during the austral spring (Berkman, Waller & Alexander, 1991), while in TNB (middle Ross Sea, ∼74°S), based on gonadosomatic index data, maturation and spawning were inferred to take place only in late summer (Cattaneo-Vietti, Chiantore & Albertelli, 1997). At Rothera Station (Antarctic Peninsula), post-metamorphic juveniles (i.e., with few growth lines after the prodissoconch II) were observed on settlement panels between February and May (Bowden, 2005). Our data, based on the direct evidence of new recruits whose age and time of settlement were calculated from different von Bertalanffy equation parameters (Fig. 6C, red and blue lines), indicate that spawning does have to occur before the summer as first settlers arrive already during the Austral spring and there is a major settlement peak in January–February, even in TNB. This timing agrees with reports of Berkman, Waller & Alexander (1991) and Dayton & Oliver (1977), who observed “thousands of recently metamorphosed (pinhead size) pectens, A. colbecki” in the sediment at Explorers Cove during the austral summer.

The mismatch with A. colbecki gonadosomatic index data previously collected in TNB (Cattaneo-Vietti, Chiantore & Albertelli, 1997) is therefore intriguing and requires further investigation. It could be tentatively explained with the possible existence of high interannual fluctuations in the timing of gamete production, in turn influenced by variability in their water-borne food supply (Chiantore et al., 2000). Indeed, A. colbecki appears to exhibit intermittent, greater-than-annual recruitment In TNB. In some years, there has been total absence of the smallest size classes, and there are large oscillations in the size-frequency distribution of individuals <30 mm (Chiantore et al., 2000; Heilmayer et al., 2003). However, these data should be treated with caution as they were based on counts of juveniles that were attached by byssus to adults (Stockton, 1984, Berkman, 1988, 1990), that had been collected using destructive gears, that is, Charcot–Picard or naturalist dredges (Chiantore et al., 2000). These counts could be potentially severely biased toward larger size classes for a variety of reasons. In fact, juveniles may become dislodged from adults during sampling due to friction with stones and other invertebrates and washed away during gear retrieval. Analogously, even smaller juveniles, that is, at the spat stage, may settle on sand and small pebbles, which are not retained by the dredge net. In this latter case the number of lost “pinhead size” juveniles can be very high, as reported by Dayton & Oliver (1977). A third bias may rely in the dredge nets themselves, as the cod-end mesh size was >one cm, that is, the size which roughly corresponds to the two smaller A. colbecki bin sizes in Chiantore et al. (2000) and Heilmayer et al., (2003). Overall, all the above points suggest that with this kind of sampling design based on destructive methods and the fortuitous retention of byssally-attached juveniles on adult shells, size-frequency data may be biased toward larger sizes, that is, abundance estimates of specimens smaller than one cm are not statistically reliable. Hence, the reported high persistency of large individuals (>60 mm) and the apparent absence of smaller cohorts in certain years (Heilmayer et al., 2003) could be, at least to a certain extent, an artefact.

It could be argued that even in our experiment water drag during mooring retrieval could have dislodged A. colbecki juveniles that had settled on the funnel surfaces. In fact, given the high speed that these structures reach during their retrieval (∼120 m min−1; Galgani et al., 2005), there could be an “opposite bias,” in this case toward smaller sizes, that is, those that due to their minute size experience less drag and water turbulence during mooring retrieval. However, in our recovery of mooring “D” (Figs. S1 and S2) from 1,117 m depth, the speed reached during retrieval was not enough to dislodge even vagile organisms. Byssally-attached ones, such as A. colbecki juveniles, should therefore be retained and hence estimates of their population structure statistically reliable. Regardless, the physical presence of these settled juveniles provides clear evidence that a recruitment event occurred during the 2007 austral spring-summer. Estimates of abundances of settlers on mooring structures, therefore emerges as a very effective method to study A. colbecki recruitment dynamics, without any apparent methodological bias. Ecologically speaking, this method is also preferable due to the lack of any impact of destructive gears on the fragile Antarctic coastal communities. Moreover, the use of coastal moorings in this way, with their records of environmental data (e.g., water current direction and intensity, temperature, etc.), may help in understanding the environmental settings/conditions at which a settlement event may occur.

Besides A. colbecki, other species settled on the mooring structures, such as the ectoparasite C. subcompressus and the pentacrinoid larvae of N. virilis, enabling the acquisition of previously unknown ecological data. In the first case, sequences of C. subcompressus found on mooring “L” helped to demonstrate a high genetic connectivity of this species between the Ross Sea and Weddell Sea-Antarctic Peninsula (Fassio et al., 2015). In the second case, the molecular characterization (Table S1) of the pentacrinoid larvae (Fig. 7I) showed that these belonged to N. virilis, a species very common in the area with the same deep purple color as the pentacrinoid larvae. In the past, this match was only inferred on the basis of color (Schiaparelli et al., 2007). Moreover, the presence of pentacrionid larvae of N. virilis on mooring structures far above the seafloor raises questions about N. virilis larval ecology. This species, in fact, is known as an ovoviviparous species (Messing, 1984; Haig & Rouse, 2008), whose pre-cystidean or cystidean larvae, when released from the female, should directly fall to the bottom to settle (John, 1939). Their presence on mooring structures implies that larvae can be dispersed by currents, and hence we are not dealing with true ovoviparity.

Conclusions

The data presented in this study demonstrate that a variety of biological and ecological data can be obtained from analysis of species that settle on OMs, and from others that are suitable for inclusion in experimental cages attached to OMs. This offers new perspectives of study with no supplementary costs other than those linked to mooring maintenance activity, which are generally already covered by the project under which OMs are deployed. Moorings belonging to permanent observation networks (such as the Italian one) are permanently at sea and retrieved annually for the short time necessary for instrument maintenance and battery replacement. They therefore represent natural “settlement panels” which offer, each year, the opportunity to evaluate possible changes in the abundances of settlers, in the structure of the fouling community and in the timing of settlement. A systematic study of the fauna found on the already existing international network of moorings deployed around Antarctica could be a valuable continental scale monitoring initiative that, if it included genetic characterization of settlers, could enable us to trace source populations and hence understand their dispersal routes. Therefore, OMs represent a potentially invaluable tool for monitoring the ecology and biology of dispersing fauna, an opportunity so far not recognized and “exploited” by researchers.

A further step in this envisaged network of “new generation moorings” would be the design and establishment of mesocosm experiments and standardized sampling protocols. For example, in collaboration with the Mooring Observatory in the Ross Sea (MORSea) (http://morsea.uniparthenope.it/?q=it), plans are in place to house modified Autonomous Reef Monitoring Structures (ARMS: https://www.pifsc.noaa.gov/cred/survey_methods/arms/overview.php) in cages similar to those used in the present study, so they can be attached to moorings. The multilayer structure of ARMS and, especially, the availability of a synthetic sponge in their first layer which traps meiofauna and bacteria, is an ideal way to characterize even the smallest organisms, and enhance our monitoring capabilities. This idea was presented as a new technological development to the Southern ocean observing system (SOOS) Ross Sea Working Group; we hope it will result in an important addition to the current use of OMs (e.g., for the monitoring of the Ross Sea Marine Protected Area).

Supplemental Information

Supplemental Information 1 Figure S1. von Bertalanffy growth curves with corrected Lmax.

von Bertalanffy growth functions from present and literature data. In this case L∞was set to 92 mm following Chiantore, Cattaneo-Vietti & Heilmayer (2003); see text for details.

Click here for additional data file.

Supplemental Information 2 Figure S2. Mooring “D” array in the 2012 version.

Mooring “D” was placed at a depth of 1086 m in the polynya facing the Drygalski Ice Tongue (75°07.773’S; 164°50.926’E), which is one of the most important areas for the production of High Salinity Shelf Waters (HSSW) in the Ross Sea.

Click here for additional data file.

Supplemental Information 3 Organisms collected on the Idromar Sediment Trap. 24 cup structures of the Mooring “D” after its retrieval (January 24th 2012).

(A) The sediment trap structure. (B) The hydroid Monocaulus sp. (MNA 5469) attached to the sediment bottles’ carousel. (C) The same specimen photographed in the lab immediately after collection. (D) Unidentified solitary ascidians. The specimens in the picture have not been sampled, but the same species was been collected on another occasions and is available in the MNA collections (MNA 10546). (E) Network of hydroids and empty shells of Limacina rangii (d’Orbigny, 1835). (F) The polynoid polychaete Harmothoe fuligineum(Baird, 1865) (MNA 3403). (G) Larva of an unidentified ‘ice-fish.’ (H) Another polynoid polychaete, Harmothoe sp. (MNA 3415).

Click here for additional data file.

Supplemental Information 4 List of species found on different moorings in the Ross Sea and curated at the Italian National Antarctic Museum (MNA), Section of Genoa.

Click here for additional data file.

Supplemental Information 5 Input file for Figure 5.

It reports von Bertalanffy growth function parameters from literature and present data.

Click here for additional data file.

Supplemental Information 6 Matlab code for Figure 6.

Click here for additional data file.

Supplemental Information 7 Matlab codes and files for Figure 6.

Click here for additional data file.

Supplemental Information 8 Input file for Figure 8.

Ten size classes of length of the juveniles found byssally attached to the funnels of the cages were considered. “Freq” represent the number of juveniles per size class bin.

Click here for additional data file.

Supplemental Information 9 Input file for Figure 4.

This reports the sizes (in mm) of the marked and caged A. colbecki specimens at the beginning of the experiment (H0 column) and their length after the 380 days of deployment at sea (H1). Valves were measured with a Vernier callipers (precision of 0.01 mm).

Click here for additional data file.

Supplemental Information 10 Excel Solver table for G-ModelFit with original data.

Click here for additional data file.

Supplemental Information 11 Input file for R Markdown script.

Adamussium growth data.

Click here for additional data file.

Supplemental Information 12 R scripts (in Markdown) for MS figures.

Click here for additional data file.

We are grateful to: E. Paschini and P. Penna (PNRA project 2009/B.09, MORSea), P. Picco and A. Bordone (PNRA project 2010/A4.01) for the collaboration in the collection of samples from mooring “D” and “L” during the XXVII PNRA expedition (2011–2012); G. Canduci (PNRA project 2013 AN1.02, PI Dr.sa Iole Leonori) and P. Penna for the sampling or organisms during the XXIX PNRA expedition. Barcoding sequences of N. virilis (juveniles and adults) were obtained in the framework of the PNRA project BAMBi (2010/A1.10). We are indebted with Sig. Zanzotto G. (“Giangio”) for the design and realization of the steel armature of cages. We are indebted to Dr. Donald Kobayashi for comments and suggestions on an earlier version of the MS and to Vonda Cummings (NIWA, Wellington) for English language check. The SOOS Ross Sea working group is acknowledged for the invitation and for the fruitful discussion on this topic during the Observing the Ross Sea workshop (Shanghai, China, China on September 11–13, 2017).

Additional Information and Declarations

Competing Interests

Author Contributions

Field Study Permissions

DNA Deposition

Data Availability

The authors declare that they have no competing interests.

Stefano Schiaparelli conceived and designed the experiments, performed the experiments, analyzed the data, contributed reagents/materials/analysis tools, prepared figures and/or tables, authored or reviewed drafts of the paper, approved the final draft.

Stefano Aliani conceived and designed the experiments, performed the experiments, analyzed the data, prepared figures and/or tables, approved the final draft.

The following information was supplied relating to field study approvals (i.e., approving body and any reference numbers):

Field research was approved by the PNRA (Italian National Research Antarctic Program), on behalf of the Italian Ministry of Foreign Affairs. The permit complies with the Protocol on Environmental Protection of the Antarctic Treaty, Annex II, art. 3.

The following information was supplied regarding the deposition of DNA sequences:

Genbank accession numbers: KR364820 (COI sequence of MNA3406), KR364840 (16S sequence of MNA3406), KR364834 (COI sequence of MNA6656), KR364853 (16S sequence of MNA6656), MK188153 (COI of MNA3403), MK188154 (COI of MNA3415), MK188155 (COI of MNA9160), MK188156 (COI of MNA9151), MK188157 (COI of MNA9159). Links to BOLD species pages have been provided in Table S1.

The following information was supplied regarding data availability:

CNR for data and Italian National Antarctic Museum (MNA, Section of Genoa) for the vouchers: MNA 9162 to MNA 9197; MNA 9161, MNA 9151; MNA 9153; MNA 9154; MNA 9159; MNA 9160; MNA 3403; MNA 3415; MNA 5469; MNA 10546; MNA 3403; MNA 3415.

All MNA vouchers can be seen on the MNA collection page (http://www.mna.it/ and https://steu.shinyapps.io/MNA-generale/). To access the database at http://www.mna.it/, scroll through the menu “cataloghi dei reperti” and select“organismi”.

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
