# Peer review of "Oceanographic moorings as year-round laboratories for investigating growth performance and settlement dynamics in the Antarctic scallop Adamussium colbecki (E. A. Smith, 1902)"

_PeerJ, doi:10.7717/peerj.6373_

## Round 0.1 · original submission · Major Revisions

I concur with the reviewers that there is merit in your manuscript and I am confident that, with appropriate revision, it can lead to publication. Please carefully note the recommendation to improve the use of language - this is not the task of reviewers, nor of a journal; rather it is the responsibility of the authors.

Reviewer 1 ·

Basic reporting

Please see attached

Experimental design

Please see attached

Validity of the findings

Please see attached

Additional comments

Please see attached

Annotated reviews are not available for download in order to protect the identity of reviewers who chose to remain anonymous.

·

Basic reporting

Comments bundled in attached pdf.

Experimental design

Comments bundled in attached pdf.

Validity of the findings

Comments bundled in attached pdf.

Additional comments

Comments bundled in attached pdf.

---

## Round 0.2 · Minor Revisions

Thank you for your revision. The small number of following points remain:

1. I do not feel you have appropriately responded to the review comment below. The modified paragraph (lines 279-290) still states currents were "toward northeast" when Fig. 6A clearly shows these were oriented northward. On line 286 you state that "currents were mainly from a north easterly direction" - please check that this should read "...mainly TOWARD a NORTHERLY direction", which would also then follow oceanographic convention (always direction toward, not from).

Review: The manuscript states that ...from September to December, currents had a similar pattern in direction... This statement is suggesting that currents were oriented SW and NE, as stated in the opening sentence of this paragraph. However, when looking at the September to December time frame, currents are weak and northward.
Answer: We have modified the whole paragraph Timeseries (lines 273-283) and this part was corrected

2. You have used both "Adamussium" and "A. colbecki" in the manuscript - including in the abstract. Please choose one of these and use it throughout. The latter ("A. colbecki", italicised) is the conventional usage. However, you may have good reason to use the genus name ("Adamussium") -- if this is so, please indicate this early in the manuscript (e.g., Line 113: "...the Antarctic scallop Adamussium colbecki (Smith, 1902; hereafter Adamussium)", and consider whether 'Adamussium' should be italicised in this case.

3. The Figure 6C caption introduces a result not discussed in the main text (i.e., the difference in settlement timings -- 2 days, 11 days -- from the two techniques applied. This is not appropriate. Please revise the text to include this finding, and/or remove it from the caption.

4. The Figure 8 caption states "The final fitting finite mixture distribution is shown in green (see text for details)". However, I cannot find the details in the text. Please revise the text as necessary (e.g., in the Results, a sentence describing the components?) and point me to the details.

·

Basic reporting

No additional comment

Experimental design

No additional comment

Validity of the findings

No additional comment

Additional comments

I am satisfied with the revisions based on the prior review. This manuscript is good to go in my opinion.

---

## Round 0.3 · accepted · Accept

Thank you for your diligence in revising as requested.

#